# The “Viral” Form of Polyarteritis Nodosa (PAN)—A Distinct Entity: A Case Based Review

**DOI:** 10.3390/medicina59061162

**Published:** 2023-06-16

**Authors:** Victoria Pașa, Elena Popa, Mihaela Poroch, Adriana Cosmescu, Agnes Iacinta Bacusca, Ana Maria Slanina, Alexandr Ceasovschih, Alexandra Stoica, Antoneta Petroaie, Monica Ungureanu, Andrei Emilian Popa, Raluca Ioana Avram, Cristina Bologa, Vladimir Poroch, Elena Adorata Coman

**Affiliations:** 12nd Rheumatology Department, Clinical Rehabilitation Hospital, 700661 Iasi, Romania; victoriapasa5@gmail.com; 2Faculty of Medicine, “Grigore T. Popa” University of Medicine and Pharmacy, 16 Universitatii Str., 700115 Iasi, Romania; adriana_cosmescu@yahoo.com (A.C.); bacusca.agnes@yahoo.com (A.I.B.); ana_maria_slanina@yahoo.co.uk (A.M.S.); alexandr.ceasovschih@yahoo.com (A.C.); alexandra.rotariu.stoica@gmail.com (A.S.); pantoneta@yahoo.com (A.P.); ungureanu_monica73@yahoo.com (M.U.); andreiemilianpopa@gmail.com (A.E.P.); ralucaioanaavram@gmail.com (R.I.A.); crisbologa@yahoo.com (C.B.); vladimir.poroch@umfiasi.ro (V.P.); elena.coman@umfiasi.ro (E.A.C.); 3Department of Family Medicine, Preventive Medicine and Interdisciplinary, “Grigore T. Popa” University of Medicine and Pharmacy, Universitatii Str. 16, 700115 Iasi, Romania; 42nd Internal Medicine Department, Sf. Spiridon Clinical Emergency Hospital, 700111 Iasi, Romania

**Keywords:** polyarteritis nodosa (PAN), inflammation, systemic vasculitis, multisystem complications

## Abstract

Classic polyarteritis nodosa (PAN) is a vasculitis with systemic manifestations that is characterized by inflammatory and necrotizing lesions affecting medium and small muscular arteries, most frequently at the bifurcation of the vessels. These lesions lead to the formation of microaneurysms, hemorrhaging ruptured aneurysms, thrombosis, and, consequently, ischemia or organ infarction. *Background and Objectives*: We present a complex clinical case of a patient with a late diagnosis of polyarteritis nodosa with multiorgan involvement. *Materials and Methods*: The 44-year-old patient, in an urban environment, presented on her own in the emergency room for acute ischemia phenomena and forearm and right-hand compartment syndrome, requiring surgical decompression in the Plastic Surgery Clinic. *Results*: Significant inflammatory syndrome is noted, alongside severe normocytic hypochromic iron deficiency anemia, nitrogen retention syndrome, hyperkalemia, hepatic syndrome, and immunological disturbances: absence of cANCA, pANCA, anti Scl 70 Ac, antinuclear Ac, and anti dDNA Ac, as well as a low C3 fraction of the plasmatic complement system. The morphological aspect described in the right-hand skin biopsy correlated with the clinical data supports the diagnosis of PAN. *Conclusions*: The viral form of PAN seems to be individualized as a distinct entity, requiring early, aggressive medication.

## 1. Introduction

Classic polyarteritis nodosa (PAN) is a systemic vasculitis that is characterized by necrotizing inflammatory lesions affecting medium and small muscular arteries, most frequently at the bifurcation of the vessels [1]. These lesions lead to the formation of micro aneurysms, aneurysm rupture with hemorrhaging, thrombosis, and, consequently, ischemia or organ infarction [2]. The pathogenesis of PAN is not fully understood, but it is thought to be related to an abnormal immune system response [1,3]. Compared to other systemic vasculitis that have been classified as idiopathic or are truly autoimmune, PAN has more identified triggering factors. Among these, hepatitis B virus (HVB) is the most frequent and well-characterized trigger of PAN, and the recognition of the etiopathogenic role of HVB in HVB-associated PAN has important therapeutic implications for these patients [1,4]. Along with HVB, hepatitis C virus (HVC), human immunodeficiency virus (HIV), cytomegalovirus, and parvovirus B19 are also suspected as triggering factors of PAN lesions for some patients [4].

The main clinical symptoms are weight loss, fever, peripheral neuropathy, renal, musculoskeletal, gastrointestinal, and/or cutaneous involvement, hypertension, and cardiac failure [3, H; 2015]. PAN can be a severe and life-threatening disease in the absence of proper diagnosis and treatment.

The focus of this article was to analyze the clinical characteristics, laboratory investigations, and outcomes based on a case followed in cooperation with surgery departments of a patient with a late diagnosis of polyarteritis nodosa with multiorgan involvement. 

## 2. Clinical Case

### 2.1. History of Present Illness

We present the case of a 44-year-old woman from an urban area, known to have liver cirrhosis with hepatitis B virus therapeutically neglected, a duodenal ulcer surgically treated for perforation, and upper digestive hemorrhages in her personal history, requiring blood transfusions. We also note that the patient was in the nephrology records from 2015 with advanced chronic kidney disease (CKD) that required hemodialysis every three days. She presented by herself to the emergency room for acute ischemia phenomena with forearm and right compartment syndrome (Figure 1a), requiring hospitalization for surgical decompression in the Plastic Surgery Clinic. 

The patient also complains of severe physical asthenia, bilateral edema in the lower limbs, arthralgias in the small joints of the hands, fists, elbows, and knees, bilateral ankles with limited mobility, bilateral myalgias of the lower limbs, elevated blood pressure values at home, increased abdominal volume, and weight loss of approximately 9 kg in the last 2 months, with progressive worsening of symptoms. 

### 2.2. Clinical Picture

During the clinical examination, we also observed: -The general condition was influenced-Pale skin and mucous membranes, jaundice, petechiae on the right forearm, *livedo reticularis* on the left lower limb, bilateral onycholysis (Figure 1a,b)-Arthritis of hands, knees, and ankles bilaterally with positive Celsus signs of inflammation (heat, redness, swelling, pain, and loss of function)-Absent pulse of the radial artery in the left hand and absent pulse of the posterior tibial artery and *dorsalis pedis* artery bilaterally-Rhythmic heart sounds, tachycardia with HR = 93/min, BP = 180/100 mmHg-Diminished basal vesicular murmur on the right, SpO_2_ = 95%,-Abdominal distension due to ascites, displaceable dullness on the flanks, non-tender to palpation

### 2.3. Investigations

A significant inflammatory syndrome was noted (high level of C-reactive protein—37.41 mg/dL, increased serum ferritin—651 ng/mL), as well as severe normocytic hypochromic iron deficiency anemia (hemoglobin—6.2 g/dL; iron—13 mcg/dL), nitrogen retention syndrome (creatinine clearance MDRD—6 mL/min/1.73 m^2^), hyperkalemia (8.1 mmol/L), and hepatic syndrome (hypoproteinemia—5.09 g/dL with low plasma (p) albumin—2.06 mg/dL). The immunological panel showed an absence of cytoplasmic antineutrophil cytoplasmic antibodies (c-ANCA), perinuclear antineutrophil cytoplasmic antibodies (pANCA), anti-topoisomerase I antibodies (anti Scl 70 Ab), antinuclear antibodies (ANA), and double-stranded DNA antibodies (anti dDNA Ab), along with a low C3 fraction of the plasmatic complement system.

The morphological aspect described in the right-hand skin biopsy correlated with the clinical data supports the diagnosis of polyarteritis nodosa (PAN) (Figure 2).

The morphological aspect corresponds to synovitis associated with periarteritis (Figure 3).

At the right upper limb, Doppler ultrasound detected important calcifying atheromatosis in the radial and cubital arteries, with pulsatile flow in the distal brachial and radial arteries. The internal jugular, subclavicular, and brachial veins were found to be patent. 

An anterior chest X-ray (Figure 4a) showed the accentuation of the bilateral lung pattern and horizontal heart.

### 2.4. Differential Diagnosis

The first differential diagnosis considered was cryoglobulinemia, because PAN-HBV (hepatitis B virus-associated polyarteritis nodosa) and cryoglobulinemia can exhibit similar clinical features, such as arthralgias (joint pain), skin lesions, abdominal symptoms, and kidney damage. Additionally, both conditions can present with certain laboratory findings, including anemia, liver and kidney function abnormalities, and a decrease in the C3 fraction of the complement [5,6]. Cryoglobulinemia is characterized by the presence of cryoglobulins—abnormal immunoglobulins that form aggregates in response to certain infections (such as hepatitis C) or autoimmune diseases [5,7]. In our case, hepatitis C virus infection is absent, and the presence of hepatitis B virus infection and the histopathological examination clarified the diagnosis of PAN-HVB.

In the differential diagnosis of PAN-HVB, we also have taken into account:Infectious diseases that can induce clinical manifestations similar to PAN-HVB or that can produce vascular inflammation: infectious endocarditis or other infections that evolve with bacteremia, mycotic aneurysm with distal embolism, hepatitis C virus infection, or HIV infection [6,7]. The screening of these infections is important in the differential diagnosis of different forms of vasculitis.ANCA-associated vasculitis: This includes granulomatosis with polyangiitis (Wegener’s granulomatosis), microscopic polyangiitis, and eosinophilic granulomatosis with polyangiitis (Churg-Strauss syndrome). These can mimic the symptoms of PAN. The distinction between these different forms of vasculitis is essential for the appropriate treatment, and in our case, the differential diagnosis was performed through laboratory determinations (absence of ANCA antibodies) and histopathological examination [4,6].Other forms of vasculitis: There are various types of vasculitis that can present with similar clinical features, such as giant cell arteritis or Takayasu arteritis. The distinction between these different forms of vasculitis is essential for the appropriate treatment, and in our case, the differential diagnosis was performed through laboratory determinations (absence of ANCA antibodies) and histopathological examination [6,8,9].Connective tissue diseases: Conditions such as systemic lupus erythematosus (SLE), rheumatoid arthritis, and Sjögren’s syndrome can present with clinical manifestations of systemic vasculitis. Behçet’s disease is another disorder we need to consider in the differential diagnosis of PAN-HVB. It is a chronic inflammatory condition characterized by recurrent oral and genital ulcers, skin lesions, eye inflammation, and various systemic manifestations. Although it primarily affects the mucous membranes, Behçet’s disease can also involve blood vessels, leading to vasculitis that can resemble PAN-HVB [6,8,9].The paraclinical data excluded the presence of these conditions in our patient.Drug-induced vasculitis: Certain medications, such as certain antibiotics, anticonvulsants, and nonsteroidal anti-inflammatory drugs (NSAIDs), can cause drug-induced vasculitis that mimics the symptoms of polyarteritis nodosa [8,9].Malignancies: In some cases, certain types of cancer, including lymphomas and leukemia, can have vasculitis-like symptoms. The possibility of an underlying malignancy was eliminated by the performed explorations [6,8].Other non-vasculitic conditions: Conditions such as atherosclerosis and embolic disease can lead to arterial occlusion and present with symptoms similar to polyarteritis nodosa. Consideration of these non-vasculitic causes is necessary for an accurate diagnosis [6,9].

HBV-PAN must be differentiated from idiopathic PAN, an important fact for the evolution and treatment of the patient. Idiopathic PAN refers to cases where the cause of the disease is unknown. It is a systemic autoimmune condition where inflammation occurs in small and medium-sized blood vessels, leading to organ and tissue damage. Idiopathic PAN is not associated with any specific infection, including the hepatitis B virus. On the other hand, HBV-PAN is a specific form of PAN that is associated with active hepatitis B virus infection. In this case, vascular inflammation is triggered by HBV infection [2,6,7].

### 2.5. Treatment and Evolution

Analgesics, antibiotics, anti-inflammatory treatment, and transfusions of erythrocyte mass were administered. In the fifth day of treatment, the patient presented multiple melena stools, apparently without an endoscopically detectable source of bleeding.

Further, Computed Tomography Angiography (Angio CT) highlights the extravasation of the contrast substance at the level of the colon (Figure 4b), and urgent surgical intervention is required, performing a right hemicolectomy. Other findings on the Angio-CT examination were dysmorphic liver, permeable portal system (portal vein diameter—15 mm); appearance of chronically damaged kidneys, small in size with left renal lithiasis; the absence of contrast of the left renal artery; liquid in hypogastrium with maximum thickness of 19 mm; multiple lumbar-aortic adenopathies on the mesenteric artery root, ileo-colic and bilateral inguinal with maximum dimensions of 15/10 mm; calcified atheromatosis of the aorta and its emergent arteries; right pleural fluid of 12 mm, “frosted glass” areas of the level of the lower right lung lobe; and a pericardial fluid layer.

The patient was discharged in a medically stationary condition, with the resolution of acute ischemia, with indications for symptomatic treatment at home and assessment after seven days. The evolution of our patient diagnosed with PAN and multiorgan involvement was unfavorable. She returned after a month with a severe general condition, dry gangrene phenomena at the level of the lower 2/3 of the left calf, and bilateral leg and right hand (Figure 5a–d). After evaluating the case, amputation of the left thigh was required. Further, after the surgical intervention and stabilization of the patient’s medical status, a referral to a specialized unit for chronic patients was recommended.

## 3. Discussion

This case is a challenge as a result of the late diagnosis with already existing multiple multisystemic complications—renal, gastroenterological, hematological, pulmonary, and cardiovascular, which can influence the patient’s short-term prognosis. This fact required a complex multidisciplinary approach regarding the clinical lab tests and imaging exploration and the establishment of therapeutic conduct even for the severe stage of evolution.

PAN was first described in 1866 by Kussmaul and Maier. During the autopsy of a patient with fever, weight loss, abdominal pain, and polyneuropathy, areas of inflammatory exudate leading to the formation of palpable nodules along medium-sized arterioles were identified [10].

PAN, like other vasculitis, affects multiple systems, although it commonly resides in the skin, joints, peripheral nerves, gut, and kidneys [6]. The lungs are not usually affected in PAN. The typical patient with PAN presents over the course of weeks or months with fever, night sweats, weight loss, skin ulcerations or painful nodules, and severe muscle and joint pain [11,12].

In order to understand PAN, it is necessary to describe how this condition was defined. From the mid-19th century through the 20th century, the term periarteritis was used to describe a spectrum of systemic vasculitis conditions characterised by the formation of arterial aneurysms and the onset of diffuse necrotizing glomerulonephritis [13,14]. The term periarteritis nodosa was changed to polyarteritis nodosa in the mid-20th century to reflect the transmural inflammation of arteries caused by this condition [15].

Further evolution in the pathogenesis of vasculitis continued in the 1980s with the discovery of anti-neutrophil cytoplasmic antibodies (ANCA). Microscopic polyangiitis (previously called microscopic polyarteritis) is a vasculitis associated with ANCA, showing features similar to those of the classical form of PAN, with additional involvement of the renal glomeruli and pulmonary capillaries [1,13,15].

### 3.1. PAN Features

In 1990, the American College of Rheumatology (ACR) established criteria in order to promote the differentiation of PAN from other forms of vasculitis. A panel of ACR physicians selected 10 pathologic features of PAN; to confirm the diagnosis of PAN, at least 3 of the 10 criteria of the ACR must be present in the conditions in which a radiological or anatomic–pathological diagnosis of vasculitis is made [16]:Weight loss of 4 kg or more,Livedo reticularis,Testicular pain/tension,Myalgia or weakness/tension in the legs,Mononeuropathy or polyneuropathy,Diastolic blood pressure greater than 90 mmHg,Elevation of urea or creatinine level not caused by dehydration or obstruction,Presence of hepatitis B surface antigen or antibodies in serum,Arteriography showing aneurysms or obstructions of the visceral arteries,Presence of polymorphonuclear neutrophils in biopsy material from a small or medium-sized artery.

The strong association between microscopic polyangiitis (MPA) and ANCA and the pathogenic and clinical differences between MPA and PAN demonstrate that these two clinical identities are most likely separate diseases [1,17]. It was not until 1994, at the Chapel Hill International Consensus Conference (CHCC), that histological criteria were developed in order to distinguish PAN from MPA based on the presence of vasculitis in arterioles, venules, and capillaries, which defines the diagnosis of MPA and excludes PAN [18].

Inflammation may begin in the intima of the vessel and progress to include the entire arterial wall, destroying the internal and external lamina, leading to fibrinoid necrosis [15]. Aneurysms develop in the weakened vessel, leading to an underlying risk of rupture and hemorrhage. Thrombi may appear at the site of the injury. As the injury progresses, proliferation of the intima or media may lead to obstruction and secondary tissue ischemia or infarction [19].

The pathogenesis of PAN is unknown and there are no available animal models for study. Hepatitis B virus (HBV) infection is strongly associated with PAN [17]. Evidence for disease induction by immune complexes is limited to HBV-related PAN, but their role in non-HBV-related PAN is unclear [1,15].

Altered endothelial cell function may be an important part of idiopathic PAN, or a consequence of it; in PAN-HVB, viral replication can lead to direct injury to the vascular wall [20]. Endothelial dysfunction can perpetuate inflammation via cytokine and adhesion molecule production [1,2,19,21]. In polyarteritis nodosa, the affected vessels become thickened and inflamed due to intimal proliferation. As a consequence, the blood vessels narrow and blood flow decreases. As a result, the affected vessels become more predisposed to thrombosis [1].

There have been few studies on biomarkers in polyarteritis nodosa [22,23,24,25,26,27]. However, anti-phosphatidylserine-prothrombin complex (anti-PSPT) antibodies could be thought of as a biomarker for the existence of PAN and therefore could support making an early diagnosis in PAN patients with cutaneous manifestations. It is assumed that prothrombin binds to apoptotic endothelial cells and phosphatidylserine, and the resulting complexes are thought to trigger the production of anti-PSPT antibodies, resulting in the development of PAN [23]. These antibodies turned out to decrease to lower levels after treatment with cyclophosphamide and glucocorticoids in comparison with the pre-treatment levels.

In addition to anti-PSPT in cutaneous manifestation of PAN, autoantibodies against lysosomal-associated membrane protein-2 (LAMP-2) [27] were also associated, but there was no significant difference when compared to values measured in healthy controls. LAMP-2 is a component of the lysosomal membrane and is believed to partake in the pathogenesis of vasculitis by binding to neutrophils, which then infiltrate small vessels of the skin [26,27].

HBV-related vasculitis almost always takes the form of PAN, which can occur at any time during acute or chronic hepatitis B virus infection, although it typically occurs within the first 6 months after infection [20].

HBV-PAN activity does not correlate with that of hepatitis, and the symptoms are identical to those of idiopathic PAN. A few small studies demonstrated that gastrointestinal manifestations, malignant hypertension, renal infarcts, and orchiepididymitis were more common in HBV-PAN [20]. The major cause of death is gastrointestinal tract involvement, which occurs in 14% to 65% of patients with PAN. The most common symptom is postprandial abdominal pain due to ischemia, and a poor prognosis is associated with it when it is transmural due to the development of necrosis of the bowel wall with perforation [28,29].

Up to 30% of PAN cases were caused by HBV infection [30]. The widespread use of the hepatitis B vaccine and the increased safety of blood transfusions has notably decreased the incidence of HBV-PAN [28], which is now reported to be less than 5% of all cases of PAN [4].

### 3.2. Genetic Associations

A wide range of inflammatory and vascular diseases, including PAN, have been connected to loss-of-function mutations in CECR1 (also known as ADA2), the gene encoding adenosine deaminase 2 (ADA2) [31,32,33,34]. In a study by Navon Elkan and colleagues [34], six families with multiple cases of systemic and cutaneous polyarteritis nodosa were identified, most with childhood onset. Recessive mutations at the CECR1 level leading to reduced ADA2 activity were found in all families [34].

Possible roles of ADA2 include regulation of activated T-cell proliferation, macrophages, and monocyte-to-macrophage differentiation. Reducing ADA2 activity may affect the adenosine inflammatory response pathway [33,34].

### 3.3. Stages

PAN is divided into subacute, acute, and chronic stages. In the subacute stage, mononuclear cell infiltration becomes prominent, while in the acute stage, polymorphonuclear neutrophils infiltrate all layers of the vascular wall [35].

In the chronic stage, the fibrinoid necrosis of the vessels causes thrombosis and tissue infarction. Aneurysmal dilatation of the involved arteries, reaching sizes of 1 cm, is a characteristic feature of PAN. Renal lesions predominantly indicate arteritis without glomerulonephritis; however, in patients with severe hypertension, glomerulosclerosis may overlap with glomerulonephritis. Pulmonary arteries are typically not involved, and bronchial arteries are rarely affected [12,36].

### 3.4. Treatment

Known as a rare disease [35], the optimal therapy for *polyarteritis nodosa* is still uncertain. The approach of treating PAN depends on the severity of disease, the presence of isolated cutaneous PAN or other isolated/mono-organic diseases, and the presence of absence of viral hepatitis [37].

Corticosteroids are the mainstay of treatment [35]. Additional schemes include [30,37]:Idiopathic PAN refractory to corticosteroids or involving major organ involvement: corticosteroids plus cyclophosphamide are standard of care;Hepatitis B-associated PAN: corticosteroids and antiviral agents (e.g., vidarabine, interferon alfa-2b) and plasmapheresis;In patients with corticosteroid-resistant or recurrent PAN, there are case reports describing response to treatment with biological agents (e.g., infliximab, etanercept, tocilizumab, tofacitinib, rituximab);Severe PAN: plasma transfusion was used.

In order to establish a treatment in PAN, a complex evaluation of the patient is necessary, including anamnestic data (medical history), detailed clinical examination, and complex paraclinical exploration. The degree of organ and system damage and the presence/absence of hepatitis B and C viruses will be investigated, and the diagnosis must be confirmed by tissue biopsy or angiography [35,36,37,38,39].

It is important for the physician to perform appropriate investigations and tests to determine the cause of PAN and differentiate between idiopathic PAN and HBV-PAN. This will guide the appropriate treatment and influence the prognosis of the disease. In the case of idiopathic PAN, treatment primarily focuses on controlling vascular inflammation through the use of immunosuppressive medications such as corticosteroids, methotrexate, or azathioprine [37,39].

HBV-PAN requires specific treatment that targets both the HBV infection and vascular inflammation. The treatment for HBV-PAN involves suppressing viral replication with specific antivirals for hepatitis B, such as lamivudine, entecavir, or tenofovir, along with the administration of immunosuppressive medications to control vascular inflammation, such as corticosteroids or other immunosuppressive agents [35,37,40].

For patients with HVB-PAN and severe disease (ulcerative or gangrenous lesions of the extremities, acute kidney damage, polyneuropathies, central nervous system lesions, mesenteric arteritis, or myocardial ischemia), Chan et al. [40] suggest the combination of antiviral agents, glucocorticoids, and plasmapheresis in addition to antiviral therapy using a nucleoside/nucleotide analog such as entecavir or tenofovir.

Glucocorticoids are known to reduce the inflammatory component of vasculitis and may improve survival in patients with HBV-PAN. However, it is important to note that they can potentially increase viral replication, which can lead to exacerbation of chronic hepatitis as a side effect [40,41].

Plasmapheresis, on the other hand, is a technique that helps remove circulating immune complexes from the blood [40,42]. This approach may be beneficial in patients with severe disease. Although the use of plasma exchange in HBV-PAN does not have strong supporting data, it is included in published treatment protocols for this condition [37,40].

Plasma exchange may be a treatment option to consider when immunosuppressive agents cannot be recommended due to severe organ damage. The presence of pathogenesis-related target molecules in serum further supports the rationale for using plasmapheresis to remove these molecules and potentially improve disease management [37,40,42]. A summary of clinical diagnosis and medication administered in HBV-PAN is presented in Table 1 [35,36,37,40,41,42].

The treatment approach for patients with HBV-related PAN with persistent symptoms who have responded inadequately to antiviral treatment or who cannot tolerate such treatment includes glucocorticoids and other immunosuppressants, following treatment protocols commonly used for patients with PAN without viral infection [37,40]. In these cases, careful monitoring of the underlying viral infection and the potential toxicity of immunosuppressive drugs is necessary.

Before starting drug treatment, in the case of glucocorticoids and other immunosuppressive drugs, patients will be tested for diabetes and infections such as latent tuberculosis and HIV [37,40]. Additionally, vaccinations will be checked and updated according to age, with the mention that live attenuated vaccines (e.g., yellow fever, chicken pox, rubella, measles, mumps vaccine) are contraindicated in immunocompromised individuals. It is also important to counsel patients on lifestyle, diet (for example, avoiding alcohol consumption), and the measures to prevent the carrier infection from HBV [37,40,42].

## 4. Conclusions and Future Perspectives

If patients are diagnosed with HBV, we should also consider the possibility of concurrent PAN if symptoms and signs of systemic vasculitis appear. In most cases, PAN is associated with “wild” HBV with high viral replication. The viral form of PAN seems to be individualized as a distinct entity, requiring early, aggressive medication [36,38].

New research efforts should prioritize the identification of new biomarkers, the development of targeted therapies, and the improvement of disease management strategies for PAN [35,39]. However, larger interventional studies are difficult to perform due to the rarity of this disease, which hinders the development of strong treatment recommendations [39].

Future perspectives for PAN may include personalized medicine, gene therapies, and new immunomodulatory drugs, thanks to advances in technology and medicine [32,33,43].

Moreover, a multidisciplinary approach to the patient with PAN must be multidisciplinary involving rheumatologists, dermatologists, neurologists, and other specialists, which is essential to provide comprehensive care to patients with polyarteritis nodosa [44,45,46].

## Figures and Tables

**Figure 1 medicina-59-01162-f001:**
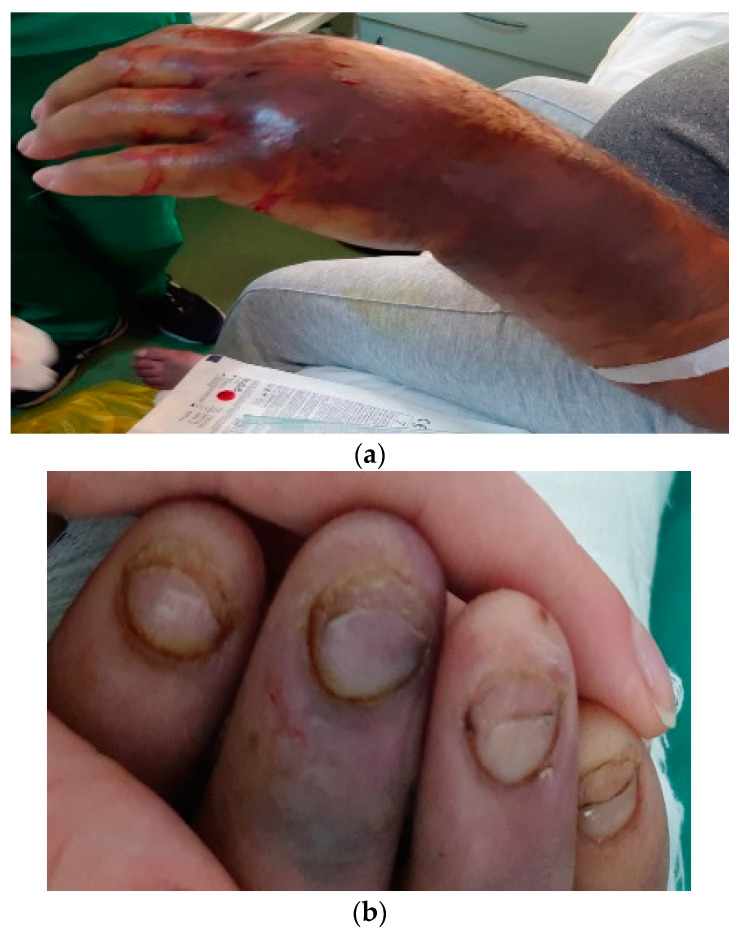
(**a**) Acute ischemia phenomena and forearm and right-hand compartment syndrome. (**b**) Bilateral hand onycholysis.

**Figure 2 medicina-59-01162-f002:**
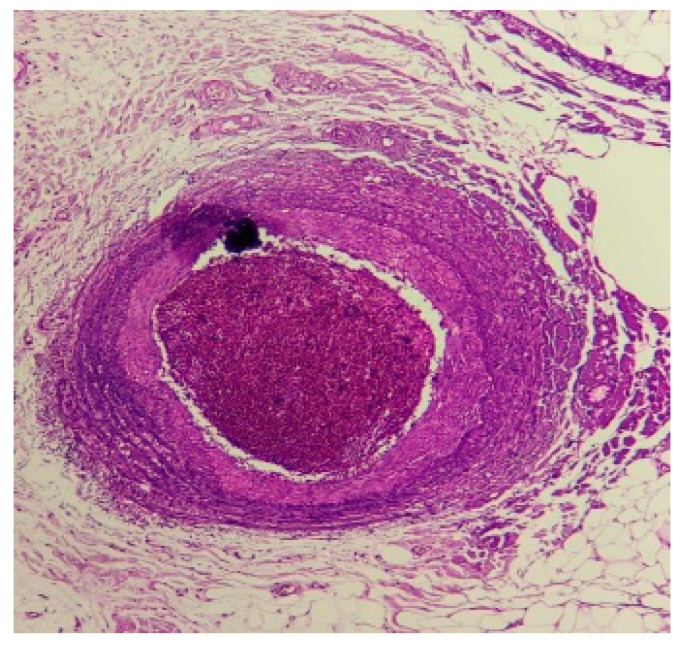
Skin biopsy (optical microscopy. Staining—hematoxylin-eosin): The dermis shows fibrosis. At the level of the hypodermis, a muscle-type artery with transmural polymorphic inflammatory infiltrate is identified, the lumen being obliterated by a recent fibrin-hematic thrombus. Fragmentation of the elastic limit.

**Figure 3 medicina-59-01162-f003:**
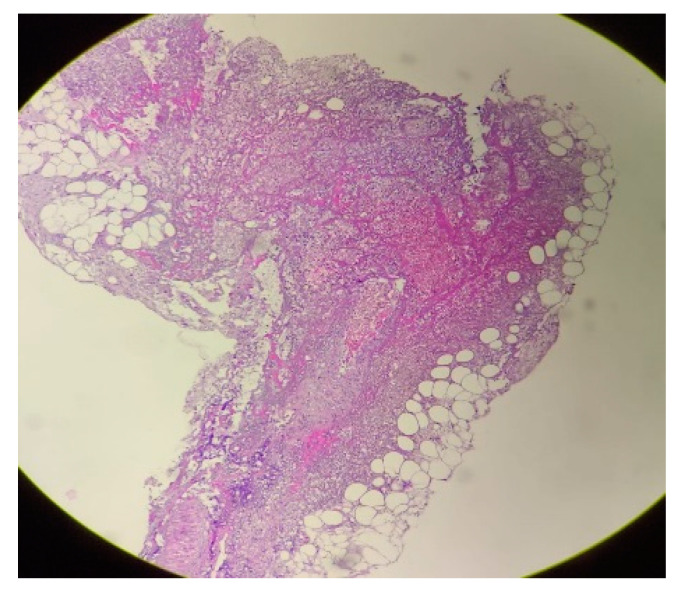
Skin biopsy (optical microscopy. Staining—hematoxylin-eosin): The synovial fragment examined is covered with a slope of synoviocytes with preserved morphology. Underneath, marked edema, congestion, lymphocytic inflammatory infiltrate, and arterial-type vessels with lumen obliterated by recent thrombi are observed.

**Figure 4 medicina-59-01162-f004:**
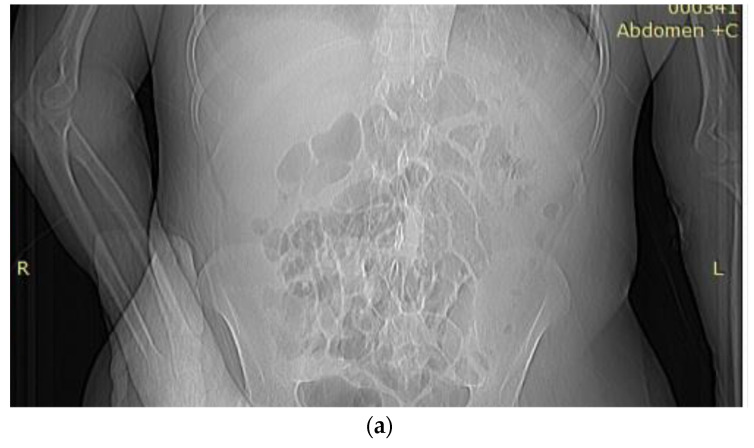
(**a**) Anterior chest X-ray showed the accentuation of the bilateral lung pattern and horizontal heart; (**b**) Angio CT highlights the extravasation of the contrast substance at the colonic level.

**Figure 5 medicina-59-01162-f005:**
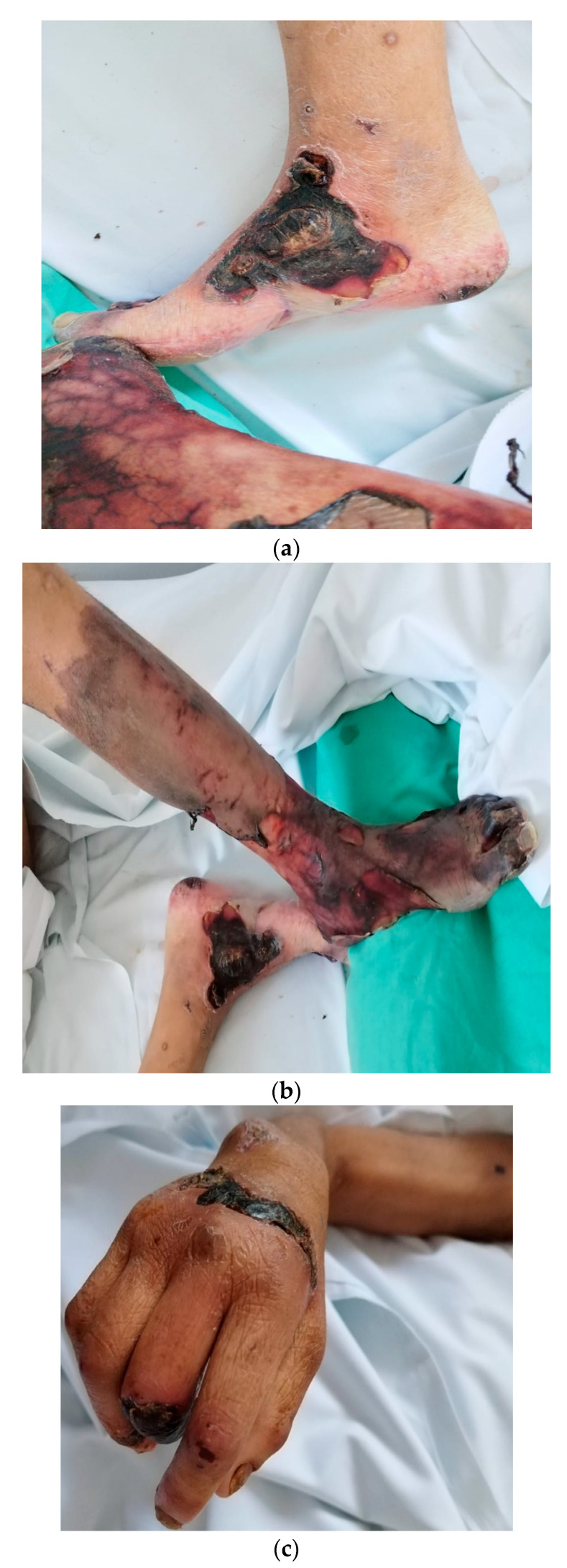
(**a**–**d**) Dry gangrene at the level of the lower 2/3 of the left calf, bilateral leg and right hand, amputation of the left thigh is performed.

**Table 1 medicina-59-01162-t001:** HVB-PAN: Summary of diagnosis and drug management.

	Clinical Features and Diagnosis	Medication
Mild Disease	- constitutional symptoms (fever, malaise, fatigue, anorexia and loss of weight, muscles aches) - arthritis - skin lesions - anemia - absence of significant cardiac, gastrointestinal, renal, or life-threatening manifestations	Antiviral therapy ^a^: - Entecavir—usually, 0.5 mg/day (for patients older 16 years; dose adjustments are necessary in cases of kidney failure) - Tenofovir: tenofovir alafenamide ^b^ (25 mg/day, aged ≥12 years) or tenofovir disoproxil fumarate ^c^ (300 mg/day, aged ≥12 years) - Short-term therapy with glucocorticoids and plasma exchange—can be instituted for patients with severe symptoms of if the disease progresses
Moderate and severe Disease	- any degree of renal failure - new or worsened hypertension secondary to PAN - symptomatic arterial stenosis - aneurysm - any ischemic diseases (e.g.: limb, cardiac, gastrointestinal, and cerebral	Antiviral therapy: - entecavir or tenofovir ^d^ - Glucocorticoids: prednisone: 0.7–1 mg/kgc/day tapered in 4 to 6 months - Plasmapheresis: 2.5–4 L/session for 6–10 sessions, daily on alternate days, over 2–3 weeks

^a^ duration of the treatment is typically for at least 12 months after HBeAg seroconversion, several years, or indefinitely; the main objective is to suppress viral replication and reduce liver inflammation caused by the hepatitis B virus. ^b^ should be avoided in pregnant women; should be avoided if the patient’s creatinine clearance (CrCl) is less than 15 mL/min and they are not on dialysis. ^c^ should be avoided if the patient’s CrCl are less than 60 mL/min. ^d^ in moderate/severe cases of HVB-PAN, the antiviral therapy is similar to the approach used in mild cases of HVB-PAN.

## Data Availability

The datasets used and/or analyzed during the current study are available from the corresponding author on reasonable request.

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
