# Peer review of "The “Viral” Form of Polyarteritis Nodosa (PAN)—A Distinct Entity: A Case Based Review"

_medicina, 2023, doi:10.3390/medicina59061162_

Round 1

Reviewer 1 Report

In this case report the authors describe a case of 44 years old presenting with acute ischemia phenomenon  right hand with  compartment syndrome. Although the diagnosis was challenging and the disease is very rare , this case has not a special unique presentation. The association with PAN e n Virus B also is well known

DIfferential diagnosi is missing.. for exaple crioglobulinemic vasculitis

For therapy , plasma exchange could be discussed

Author Response

I thank you for your review. I have made changes to the article according to your comments

Point 1: DIfferential diagnosi is missing.. for exaple crioglobulinemic vasculitis

Response 1: We have included the differential diagnosis in the article, including cryoglobulinemic vasculitis

Point 2: For therapy , plasma exchange could be discussed

Response 2: In the updated version, we completed the treatment for HBV-related PANA,  and we also discussed plasmapheresis.

Kind regards,

Elena Popa,

MD, PhD.

Reviewer 2 Report

The article is clear and well structured.

The patient's case is well presented.

However, in my opinion, the "review" section has not been deepened afterwards  , a table showing the main elements for diagnosis (divided by body organ) and treatment of the condition would be very helpful 

there is a great deal of attention to the description of the individual case but little depth to the relevant literature, an important prerogative in the publication of a review 

Minor editing of English language required

Author Response

I thank you for your review. I have made changes to the article according to your comments

Point 1: However, in my opinion, the "review" section has not been deepened afterwards  , a table showing the main elements for diagnosis (divided by body organ) and treatment of the condition would be very helpful 

Response 1: We have completed the "review" section and we have included the requested table.

Point 2: there is a great deal of attention to the description of the individual case but little depth to the relevant literature, an important prerogative in the publication of a review 

Response 2: In the updated work, we made a synthesis of the current articles in the field of HBV-PAN. We hope you consider the article good for publication.

Kind regards,

Elena Popa

MD, PhD

Round 2

Reviewer 1 Report

the authors improve substantially the manuscript, I do not have major concerns